# Design of Microstrip Antenna Integrating 24 GHz and 77 GHz Compact High-Gain Arrays

**DOI:** 10.3390/s25020481

**Published:** 2025-01-16

**Authors:** Junli Zhu, Jingping Liu

**Affiliations:** School of Electronic and Optical Engineering, Nanjing University of Science and Technology, Nanjing 210094, China; zhujunli@njust.edu.cn

**Keywords:** microstrip antenna, corner-fed, high gain, miniaturized

## Abstract

The swift advancement of contemporary communication technology, along with the development of radar systems, has raised the requirements for antenna systems. In this work, an integrated array antenna operating in the 24 GHz and 77 GHz frequency bands is proposed. The microstrip antenna array element uses a width reduction approach to reduce its volume by 39.82%. By using corner series feeding, a 3 × 3 planar array is created. The arrays operating at 77 GHz and 24 GHz can produce gains of 14.19 dBi and 15.34 dBi, respectively, with sidelobe levels of less than −9.14 dB and −12.85 dB and cross-polarization levels of −29.26 dB and −40.52 dB. This design reduces the volume of the array, eliminates the need for a complex feeding network, minimizes feeding losses, and enhances the antenna’s gain, all while maintaining good sidelobe levels and cross-polarization performance. These improvements hold significant potential for broader application. Moreover, the simulation and measurement results are in close agreement.

## 1. Introduction

With the development of modern radar systems and electronic countermeasure technologies, the number of electronic devices on radar and communication platforms continues to increase, and antennas are intertwined, which brings challenges to the multi-band operation, miniaturization, and integration of antennas. As a vital component of a radio system’s front end, the antenna has a considerable impact on overall system performance. As a result, when designing an antenna, the requirements of small size, high gain, simple structure, and dependability must be met.

Radar systems with multiple independent channels have greatly improved detection, resolution, and anti-interference ability. The cooperative multi-channel work may detect tiny targets and enhance the target angle, speed, and distance resolution. This is important for high-sensitivity, high-reliability applications. In addition, dual-band radar offers significant advantages in terms of target detection and identification performance, system anti-interference, and environmental flexibility. In contrast to 24 GHz, which has strong penetration in challenging conditions like rain, fog, smoke, and dust, 77 GHz has a shorter wavelength and better detection accuracy. As a result, dual-band radar has several applications in sectors such as mine environment awareness, weather monitoring, automotive radar, and stealth target identification. The demand for dual-band integrated antennas has thus emerged.

Microstrip antenna arrays, with their advantages of small size, low profile, light weight, and ease of integration, show great potential in radar systems, satellite communication systems, and other fields. As wireless communication technology evolves, there is a clear trend towards miniaturized and highly integrated antennas. Simultaneously, to enhance the system’s signal-to-noise ratio, the antenna’s gain must meet specific requirements. Generally, an antenna’s gain is closely related to its aperture, and miniaturization tends to reduce gain. Hence, resolving the conflict between antenna gain and size presents a significant challenge.

Currently, there are two main approaches to achieving the miniaturization of microstrip antennas. The first involves altering the parameters of electromagnetic materials, including using high-dielectric-constant substrates [1], incorporating novel artificial electromagnetic materials such as electromagnetic band gap (EBG) structures [2,3], and employing metamaterials [4]. The second approach focuses on modifying the structure of the radiating elements or the ground plane to increase the current path length, thereby achieving smaller patch sizes. This can be achieved through techniques such as meander line fractal technology [5,6,7], folded patches [8], surface slotting [6,9,10,11,12,13], and loading methods [14]. This paper proposes reducing the width of the patch to achieve antenna miniaturization. Compared with loading artificial electromagnetic structures, reducing the patch width can not only effectively achieve the purpose of miniaturization, but also has less impact on the radiation characteristics of the antenna, especially when performing corner feeding, which can greatly reduce cross-polarization. This is very important for maintaining antenna performance.

In antenna design, a key objective is to minimize the antenna’s size while maximizing its performance. One of the most critical performance metrics is antenna gain. Currently, there are several methods to enhance the gain of microstrip antennas: adding electromagnetic band gap (EBG) structures to the substrate to suppress surface waves and alter the internal field distribution of the antenna [15,16]; introducing EBG structures or loading Fabry–Perot resonators on the overlay to create reflective structures that increase in-band gain [17,18]; and loading low-refractive-index structures [19,20]. Among these methods, adding EBG structures to the substrate provides limited gain improvement and requires precise manufacturing techniques, making it less suitable for high-frequency applications. The other two methods involve adding electromagnetic structures at a certain height above the antenna surface, significantly increasing the antenna’s profile height, which is detrimental to miniaturization efforts. In practice, the feed network also plays a crucial role in determining the antenna’s gain performance. In the design presented in this paper, we employ a corner-fed method between rectangular patches, connecting the diagonally opposite corners of adjacent microstrip radiating elements. This approach significantly reduces the complexity of the feed network and adds an energy servo path to the patch unit. Additionally, compared to parallel feed networks, the angular feed method shortens the length of the feed lines, thereby reducing energy loss along the feed lines and enabling the antenna array to achieve higher gain.

This paper presents the design of an integrated antenna array characterized by its simple structure, compact size, and considerable gain. By reducing the width of the microstrip radiating elements, the volume of each antenna unit is minimized. The array utilizes a corner-series-fed method, which enhances the antenna array’s gain and achieves satisfactory cross-polarization levels and sidelobe levels. The antenna array design methodology is applied to both the 77 GHz and 24 GHz frequency bands, integrating them onto a single dielectric substrate. The design of a small, four-channel, high-gain integrated array antenna is accomplished while ensuring good isolation between the channels.

## 2. Antenna Unit

Rectangular microstrip patch antennas are usually designed based on transmission line models and their derived empirical equations. The result of the calculation is the patch size operating in the TM01 mode state. The antenna design dimensions can be calculated via the following four steps:
Calculate the patch width: $W=\frac{c}{2f}{\left(\frac{\varepsilon_{r}+1}{2}\right)}^{-\frac{1}{2}}$.Calculate the equivalent dielectric constant of the patch: $\varepsilon_{e}=\frac{\varepsilon_{r}-1}{2}(1+\frac{10h}{W}{)}^{-\frac{1}{2}}+\frac{\varepsilon_{r}+1}{2}$.Calculate the extension length of the patch Δl: $\Delta l=0.412h\frac{\left(W/h+0.264)(\varepsilon_{e}+0.3\right)}{\left(W/h+0.8)(\varepsilon_{e}-0.258\right)}$.Calculate the length of the patch: $L=\frac{c}{2f\sqrt{\varepsilon_{e}}}-2\Delta l$.
where *c* is the speed of light, *ε_r_* is the relative dielectric constant, *f* is the operating frequency, and *h* is the thickness of the dielectric substrate.

In this chapter, a microstrip patch antenna design is presented with a center frequency of 77 GHz, utilizing a Rogers 5880 dielectric substrate with parameters $\varepsilon_{r}=2.2$ and tanδ = 0.0009 and a thickness of 0.254 mm. Based on the given formulas, the theoretical patch width (*w*) is calculated to be 1.54 mm, and the patch length (*l*) is 1.14 mm. Figure 1a illustrates the simulation model of the antenna element. Electromagnetic simulation and optimization indicate that the patch resonates at 77 GHz when the width is 1.14 mm and the length is 1.12 mm. The primary polarization component of the antenna’s radiation pattern is mainly dependent on the electric field radiation from the narrow edge, while the radiation from the longer edge constitutes the cross-polarization component, where the narrow side is the side with length *w* and *l* is the size of the long side. In the TM01 mode, the long edge contributes negligibly to the radiation, resulting in minimal cross-polarization components when the antenna operates in the fundamental mode. However, there will be some adjustments if the antenna element’s feeding location is moved to the corner. The antenna element will operate with the sum of two almost equal amplitude modes, namely TM01 and TM10 modes, resulting in two mutually orthogonal polarizations because the patch’s length and breadth are comparable.

To lower the volume of the array and avoid orthogonal mode excitation in the case of corner feeding, we may begin with the antenna unit by correctly reducing its width and modifying the length of the radiation patch so that it resonates at 77 GHz. The antenna unit with reduced width mainly works in TM01 mode. The cross-polarization component theoretically stays unaltered as the reduced-width patch element does not alter the operating mode under the condition of center feeding. On the other hand, the radiated energy theoretically drops with the narrowed edge lessened. Moreover, the reduced-width patch impedance rises since the patch’s impedance is directly correlated with its thin edge size *w*, which could have an impact on the antenna’s bandwidth. The optimized miniaturized antenna element model is displayed in Figure 1b. The width of the miniaturized element is 0.68 mm, and the length is 1.13 mm. Compared to the initial antenna element, the volume of the miniaturized element is reduced by 39.82%.

Figure 2 and Figure 3 present the simulation results of the antenna element. As shown in Figure 2, before miniaturization, the minimum point of S11 is at 77.10 GHz, with a value of −44.65 dB. The antenna covers a bandwidth of 4.04 GHz, ranging from 75.10 GHz to 79.14 GHz. After miniaturization, the minimum point of S11 shifts to 76.96 GHz, with a value of −33.99 dB. The antenna bandwidth is now 3.18 GHz, spanning from 75.39 GHz to 78.57 GHz. Both perform well in return loss at 77 GHz. Compared to the pre-miniaturized state, the relative bandwidth of the antenna is reduced from 5.23% to 4.13%, but the bandwidth reduction of 1.1% is within an acceptable range.

Figure 3a shows the simulated radiation patterns of the antenna element. Before miniaturization, the antenna’s maximum gain is 7.43 dBi, with a half-power beam width of 76.6°, and cross-polarization at θ = 0° is −51.68 dB. After miniaturization, the antenna’s maximum gain is 6.61 dBi, with a half-power beam width of 73.65°, and cross-polarization is −52.16 dB. Compared to the initial antenna element, the impact of miniaturization on cross-polarization levels is minimal in the case of center feeding, and there is only a slight difference in gain. The radiation patterns of the two patches that receive corner feeding are displayed in Figure 3b. The figure shows that the initial patch’s cross-polarization under corner feeding is −5.16 dB, while the patch’s cross-polarization after width reduction is −21.03 dB. Following downsizing, the patch’s cross-polarization is significantly decreased.

From the above simulation results, it can be concluded that reducing the patch width effectively achieves the goal of minimizing the antenna size. This method has a negligible impact on the antenna’s bandwidth and gain and greatly reduces the cross-polarization ratio in the case of corner feeding, making it a reliable approach for antenna element miniaturization.

## 3. Array Antenna Structure

### 3.1. The 77 GHz Antenna Array

The design of the dual corner-fed high-gain antenna array presented in this paper is illustrated in Figure 4. Consistent with the design of the individual antenna elements, the antenna array utilizes a dielectric substrate with a thickness *h* of 0.254 mm and a relative permittivity *ε_r_*of 2.2. The back side of the antenna features a metallic ground plane, while the front side has metallic radiating patches. The rectangular patches serve as the radiating elements of the array, and adjacent patches are interconnected at their corners using microstrip lines to facilitate feeding between them. A 50 Ω coaxial connector feeds the array from the backside of the central element.

The 1 × 3 linear array consists of three elements, with the entire array fed by a coaxial feed located at the center of one of the radiation edges of the central element. Microstrip lines are added diagonally from the corner of the rectangular patch, where the radiation edge is located, to the diagonally opposite corner of the adjacent element to achieve feeding of the adjacent patch. The ends of the linear array are symmetrically distributed. By translating the linear array upwards and downwards and connecting adjacent patches with feed lines, a 3×3 planar array is obtained. The planar antenna array is composed of nine elements. The radiation edges and non-radiation edges of each rectangular patch are connected to the corresponding edges of the adjacent patches, with the corner feed line length being approximately half of the guided wavelength *λ_g_*/2, thereby achieving excitation of each patch with the same phase. Simultaneously, all patches display constant transverse and longitudinal spacing, with the feed point situated in the array’s center.

The 1 × 3 linear array is formed along the H-plane direction (y-axis direction) of the patch. Along the feed line, the energy travels to the patch’s corner, where it can excite the internal TM01 mode and produce a strong radiation effect. This feeding technique minimizes the space taken up by the feeding network and simplifies the array feeding network in the H-plane direction. In addition, it can decrease the energy loss brought on by the complicated feeding network and carry out effective energy transfer.

The planar array is corner-fed in the E-plane direction of the patch. Each element in this scenario has two options for obtaining energy: feeding along the x-axis and feeding along the y-axis. Naturally, the path to the patch from the array’s center feed point might not be unique, particularly if the array is increased. Nevertheless, some energy will be radiated by the current flowing through the patch, causing various energies to arrive at the patch via various routes. The field in the patch excited by the path that consumes the least energy is dominant, and the energy field brought by the direction perpendicular to it is superimposed on the main field. After that, the current connects to the patch behind and transfers the energy via other feeding points.

The corner series feeding approach boosts the energy of each array element, enhances the antenna’s feeding efficiency, and gives the array element a new energy receiving direction as compared to the conventional array feeding method that combines series and parallel feeding. Additionally, the array element located farthest from the feeding center will acquire less energy, demonstrating the energy gathering characteristic toward the center. As a result, the array may become more delicate, the antenna size may decrease, the beam may become more focused, and the antenna gain may increase.

The length of the feeder will be directly impacted during the design process by the separation between the radiating edge and the non-radiating edge of neighboring elements, which will impact the elements’ feeding phase. Because the feeder’s width and impedance are directly correlated, adjusting the feeder’s width will also alter the array’s impedance matching. There may also be an impact on the antenna’s electromagnetic performance, including the return loss and pattern.

The 1 × 3 linear array model, which was obtained through simulation optimization, has the following overall dimensions: the feed line width *wy* is 0.15 mm, and the array element spacing *ly* is 0.85 mm. The simulated linear array pattern is shown in Figure 5. The model’s *xoz* plane simulation results are correlated with the antenna pattern in the E-plane, and the *yoz* plane results are coupled with the pattern in the H-plane. The graphic illustrates that the antenna’s highest gain, which is 10.10 dBi with a sidelobe level of −14.29 dB, lies in the Z-axis direction. The H-plane’s 3 dB beam width covers 42.20 degrees, ranging from −20.90 degrees to 21.28 degrees. The cross-polarization of the linear array at *θ* = 0 deg is −53.54 dB.

In the 3 × 3 planar array, the distance between the nearest radiating edges of two adjacent linear arrays is denoted as *lx*. After simulation and optimization, the longitudinal spacing *lx* of the elements is determined to be 1.6 mm, and the lateral spacing *ly* is 0.85 mm. The widths of the feed lines connecting the radiating edges laterally and longitudinally, termed *wy* and *wx*, are both set to 0.15 mm. As shown in Figure 4, a short stub is added to the patch where the coaxial feed is positioned. The length of the short stub *lm* is 0.4 mm, and its width *wm* is 0.2 mm. This adjustment is necessary due to the increased number of patches, which causes energy to concentrate towards the center, leading to an induced load that challenges the impedance matching of the antenna. By varying the length and width of the short stub, the reflection phase can be adjusted, thereby improving the return loss characteristics of the antenna, which is a practical solution in engineering applications.

Figure 6 presents the simulated radiation pattern of the planar array. The maximum gain of the array is 14.19 dBi, with the E-plane beam width being 22.65 degrees and the sidelobe level at −9.14 dB. The H-plane beam width is 45.48 degrees. The cross-polarization level of the array is −29.26 dB. Compared to the 1 × 3 linear array, there is a noticeable increase in the cross-polarization of the antenna. The primary cause of this phenomenon is the use of diagonal feeding along the radiation edges. On one hand, feeding from the corner makes the internal field distribution within the patch asymmetrical compared to the central feeding. The energy near the feeding point is always slightly larger, leading to an imbalance in the distribution of electric fields, magnetic fields, and currents at the edges of the patch. Their magnitude and direction can directly affect the radiation performance of the antenna. The asymmetry of the field distribution in the corner-fed patch is the main reason for the degradation of its cross-polarization performance. In addition, the diagonal feed lines connecting the radiation edges of adjacent patches also damage the cross-polarization performance of the array. According to the transmission line model of the patch, there is a discontinuity between the patch unit and the microstrip transmission line in the direction of the radiation edge, with a radiation gap present. Adding a feed line at the end of the radiation gap blocks the radiation gap at one corner of the patch, which exacerbates the asymmetry of the patch’s field distribution. The diagonal feed line connecting the two radiation edges also generates radiation at the corner. Since the feed line is diagonal, there is an angular difference between the radiation field it generates and the radiation field produced by the patch, which also weakens the cross-polarization of the array antenna. Although the cross-polarization of the antenna has increased, it still meets the general engineering design requirements.

The simulated return loss parameters of the antenna array can be observed in Figure 7. Without the use of a matching stub, the antenna exhibits significant return loss. After employing a stub, the antenna’s bandwidth covers a 3.15 GHz frequency band ranging from 75.62 GHz to 78.77 GHz.

### 3.2. The 24 GHz Antenna Array

We utilized the same methodology as for the 77 GHz array when developing the antenna array centered at 24 GHz. The dielectric substrate used in the design process for the 24 GHz frequency band is consistent with that of the 77 GHz frequency band, using a Rogers 5880 substrate with a thickness *h* of 0.254 mm and a relative permittivity *ε_r_* of 2.2 to facilitate the integration of antenna arrays operating at two different frequency bands. Figure 8 displays the 24 GHz antenna array model. A coaxial feed from the backside of the patch powers the array, and the feed point is situated at the edge of the patch unit in the middle of the array. Similarly, to negate the reactive load of the array and enhance its return loss characteristics, a short-circuiting stub is inserted next to the feed point. Corner feeding is used to connect the array’s constituent components.

The antenna model’s dimensions were established via simulation and optimization. The patch length *l* is 3.89 mm, while the patch width *w* is 2.78 mm. The array’s feed line width *wy* is 0.43 mm, and its lateral element spacing *ly* is 1.79 mm. The feed line width *wx* is 0.30 mm, as the longitudinal element spacing *lx* is 4.60 mm. The dimensions of the short stub are 0.2 mm in width and 1.2 mm in length *lm*.

Figure 9 shows the simulated radiation pattern of the 24 GHz planar array. The array has a maximum gain of 15.34 dBi, a sidelobe level of −12.85 dB, and an E-plane beam width of 26.43 degrees. The beam width of the H-plane is 36.60 degrees. The array has a cross-polarization level of −40.52 dB. The simulated outcomes of the array’s return loss are shown in Figure 10. At 23.94 GHz, the antenna’s S11 strikes its lowest value of −39.98 dB. The antenna’s S11 at 24 GHz is −20.22 dB. The antenna’s relative bandwidth is 1.58%, with its absolute bandwidth spanning 0.38 GHz between 23.76 GHz and 24.14 GHz.

Table 1 lists the dimensions and electromagnetic performance of the antenna designed in this article and compares them with the existing related literature. The array layout approach in reference [21] is constrained and can only feed at the center of the prototype patch, as Table 1 illustrates. Furthermore, there are no clear benefits to the cross-polarization and gain over this article. The sidelobes of the approach suggested in reference [22] are quite high, and its relative bandwidth is just 2.4%. The beam in reference [23] is clearly deflected and does not point in the patch’s normal direction. The parallel feeding technique is used in Reference [24] to create a linear array. There is a big array area since the feeding network takes up a lot of room. The back-coupled series feeding technique is used in Reference [25]. Integration is hampered by the antenna’s inability to be directly affixed to other items’ surfaces. Furthermore, if the antenna’s E-plane and H-plane beam widths must be met, a planar array combination is necessary, and simple series or parallel feeding is insufficient. Using the series and parallel mixed feeding approach will result in an increase in the antenna’s volume. Space is wasted and gain is lost because more array components cannot be arranged in a smaller area and obtain a higher gain. As a result, the technique of combining corner feeding and reduced patch technology to create a planar array can utilize the entire array space and offers exceptional performance in terms of high gain and miniaturization. Its sidelobe, bandwidth, cross-polarization, and other characteristics are also excellent. As a result, the method described in this article of combining reduced patch technology and corner feeding technology to form a planar array can make full use of the array space and has excellent performance in terms of high gain and miniaturization, as well as cross-polarization, bandwidth, sidelobe, and other characteristics.

### 3.3. Integrated Array

We further developed the integrated array design after creating antenna arrays for two distinct frequency bands. The integrated array, as depicted in Figure 11, consists of two 24 GHz arrays and two 77 GHz arrays merged on a 25 mm radius dielectric substrate. The integrated array continues to employ a Rogers 5880 dielectric substrate with a thickness h of 0.254 mm, in accordance with the design mentioned above. Along two mutually perpendicular diameters, the two groups of arrays for distinct frequency bands are symmetrically distributed.

The four separate antenna arrays that make up the planned integrated array equate to a four-port device. As signal interference in a radar system can have a significant impact on the system’s stability and dependability, there are strict criteria for the isolation of each port. The larger the distance between the arrays, the better the isolation will be between each port. However, if the spacing between the arrays is too great, the edge of the array will be closer to the edge of the dielectric substrate, which will also affect the antenna’s electromagnetic performance. As a result, the array’s configuration and the choice of array spacing are critical. The results of the simulation show that there are 28 mm separating the two 24 GHz planar arrays and 34.59 mm between the two 77 GHz planar arrays. In this instance, Figure 12 displays the simulation results of the isolation between the arrays.

Because the two groups of antennas are perfectly symmetrical, the array can be considered a reciprocal network, with the forward transmission coefficient between the ports equal to the reverse transmission coefficient. Therefore, the isolation between the arrays can be fully characterized by the one-way transmission coefficient in the figure. Figure 12a displays the energy transfer when a 24 GHz antenna acts as a sending antenna and the remaining three antennas act as receiving antennas. The antenna’s isolation in the entire frequency spectrum near 24 GHz exceeds 20 dB, while the isolation inside the antenna bandwidth exceeds 50 dB. Figure 12b shows the energy transmission when a 77 GHz antenna is used as a transmitting antenna. The isolation between antennas is more than 30 dB throughout the whole frequency range around 77 GHz, while the isolation within the bandwidth is more than 35 dB. This demonstrates that the design satisfies the need for each antenna in the integrated array to be independent.

We processed and measured the simulated antenna. In accordance with the design, a Rogers5880 plate measuring 0.254 mm in thickness was chosen as the dielectric substrate. At the feed point, a suitable microwave coaxial connector was chosen for welding. A picture of the constructed integrated array is shown in Figure 13. The measured antenna bandwidth covers a total bandwidth of 2.5 GHz between 76.65 GHz and 79.15 GHz. Compared with the simulation, the lowest value of S11 is shifted to the right by 0.65 GHz, and the bandwidth is reduced. Figure 14 shows a comparison of the S11 simulation and measurement of the 77 GHz antenna in the integrated array. A comparison of the simulation and the measurement of the antenna gain at 77 GHz is shown in Figure 15. The measured antenna gain is 13.46 dBi, the half-power beam width of the E-plane is 23.21, the sidelobe level is −5.66 dB, and the half-power beam width of the H-plane is 32.64. The simulation pattern and the measured data agree well.

A comparison of the measurement of the 24 GHz antenna in the integrated array and the S11 simulation is shown in Figure 16. The measured antenna bandwidth spans 0.5 GHz, which is the entire frequency range between 23.91 GHz and 24.41 GHz. At 24.18 GHz, which is 0.75% off from the design value, S11 has its lowest value. The measured absolute bandwidth of the antenna is slightly wider than the results of the simulation. The two curves have comparable tendencies in the 20–27 GHz frequency range, with the measured curve’s minimum point being slightly higher than the simulation curve. A comparison of the simulation and measurement gain of the 24 GHz antenna in the integrated array is shown in Figure 17. The measured antenna has a maximum gain of 15.23 dBi, an E-plane half-power beam width of 22.0°, a sidelobe level of −11.67 dB, an H-plane half-power beam width of 39.0°, and a sidelobe level of −20.10 dB. There is agreement between the modeled and measured radiation patterns.

## 4. Conclusions

In this work, an integrated array antenna with two small, high-gain planar arrays operating at 77 GHz and two at 24 GHz is designed. The antenna unit has a narrower, more compact design. With the 77 GHz antenna unit, for instance, the volume of the reduced unit is decreased by 39.82% when compared to the original patch design, while the performance of gain, bandwidth, cross-polarization, etc., is kept at an exceptionally high level. With the small patch serving as the unit, a corner series feeding technique is used to produce a 1 × 3 linear array in the non-radiating edge direction. This results in a gain of 10.10 dBi at 77 GHz, with the sidelobe level and cross-polarization being less than −14.29 dB and −53.54 dB, respectively. A corner-fed planar array is produced by expanding the linear array in the direction of the radiating edge and adding a short stub at the coaxial feeding for matching in order to increase the antenna gain. A 14.19 dBi gain of the 77 GHz planar array is achieved, and the corresponding sidelobe level and cross-polarization are −9.14 dB and −29.26 dB. The 24 GHz array has a gain of 15.34 dBi, a sidelobe level of −12.85 dB, and a cross-polarization of −40.52 dB. To create a four-element integrated array antenna, the two sets of arrays mentioned above in the various frequency bands are merged on a dielectric substrate with a radius of 25 mm. There is sufficient separation between the antennas. After processing and measuring the integrated array, the outcomes largely agreed with the simulation, indicating good performance.

## Figures and Tables

**Figure 1 sensors-25-00481-f001:**
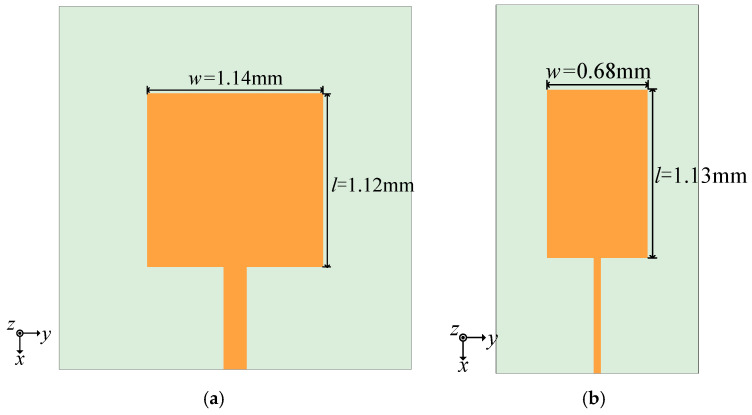
(**a**) Antenna unit model. (**b**) Miniaturized antenna unit model.

**Figure 2 sensors-25-00481-f002:**
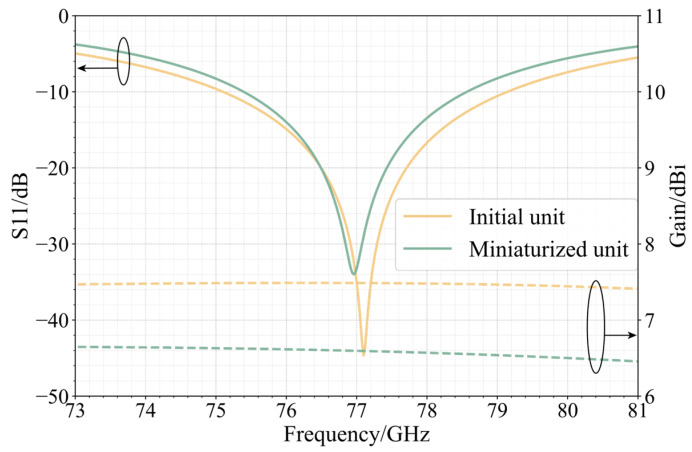
S11 simulation results and gain curve of antenna elements.

**Figure 3 sensors-25-00481-f003:**
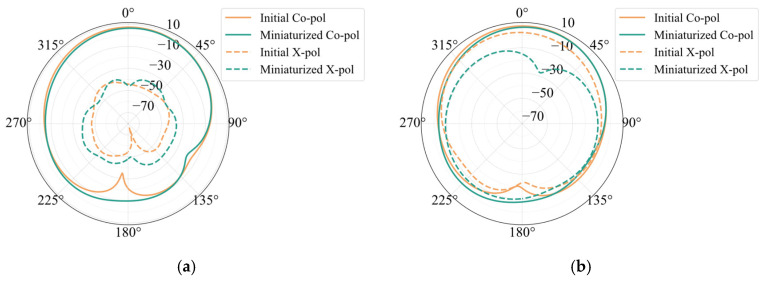
Simulation results of antenna element pattern: (**a**) center feed and (**b**) corner feed.

**Figure 4 sensors-25-00481-f004:**
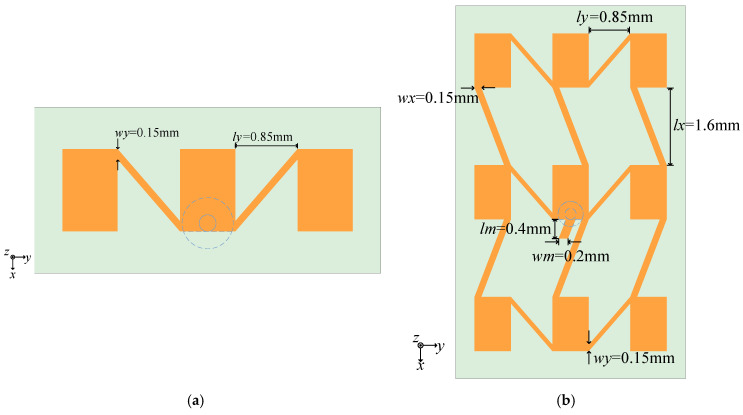
Array antenna structure: (**a**) 1 × 3 line array and (**b**) 3 × 3 area array.

**Figure 5 sensors-25-00481-f005:**
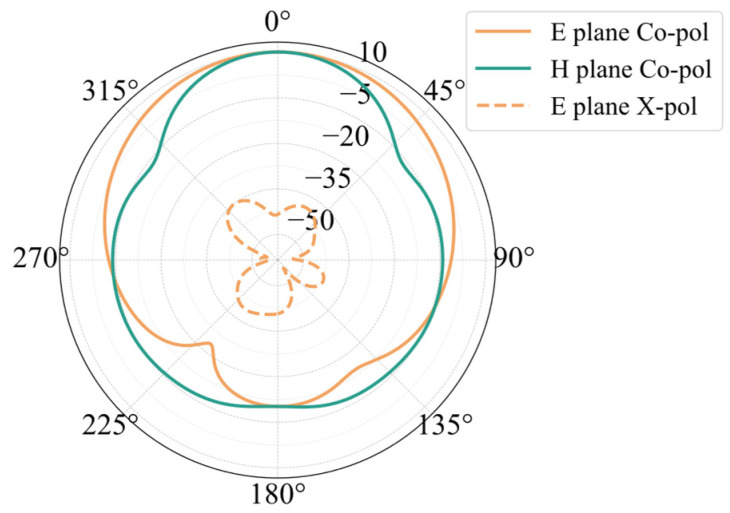
Simulation results of 1 × 3 linear array pattern.

**Figure 6 sensors-25-00481-f006:**
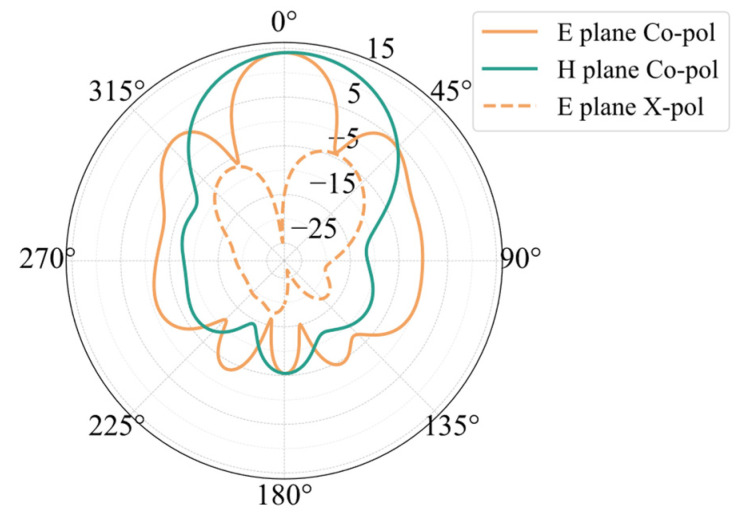
Simulation results of planar array pattern.

**Figure 7 sensors-25-00481-f007:**
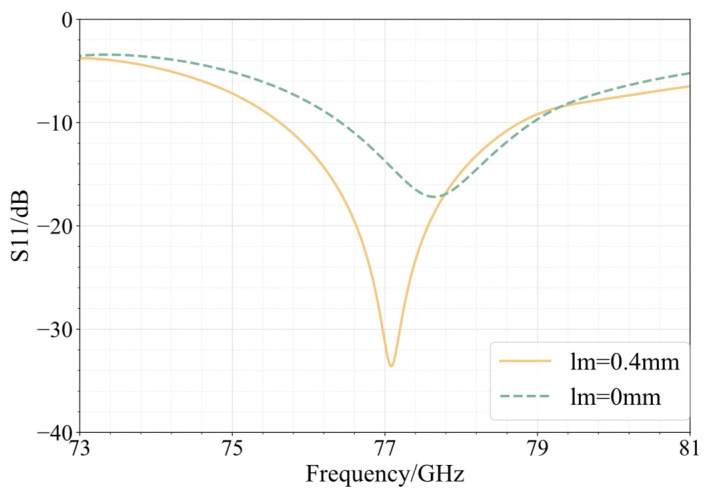
S11 of planar array.

**Figure 8 sensors-25-00481-f008:**
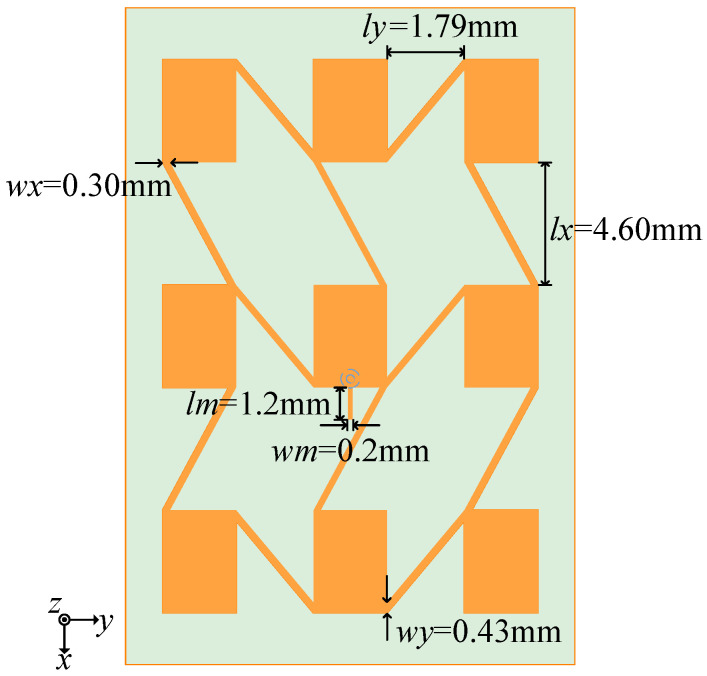
Structure of 24 GHz antenna.

**Figure 9 sensors-25-00481-f009:**
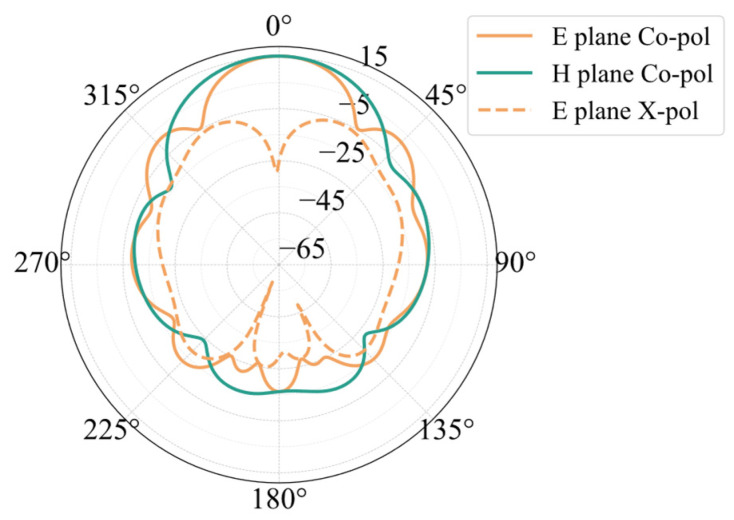
The 24 GHz array radiation pattern simulation results.

**Figure 10 sensors-25-00481-f010:**
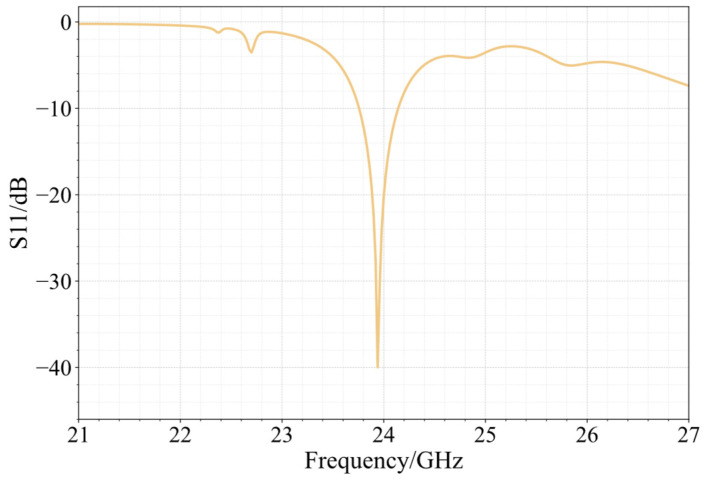
The 24 GHz array S11 simulation results.

**Figure 11 sensors-25-00481-f011:**
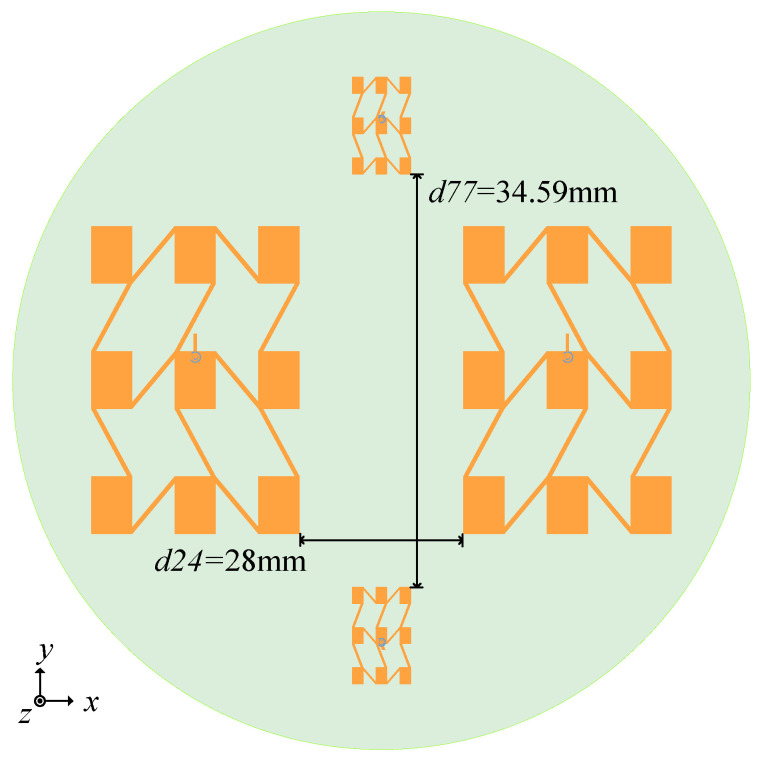
Integrated array model.

**Figure 12 sensors-25-00481-f012:**
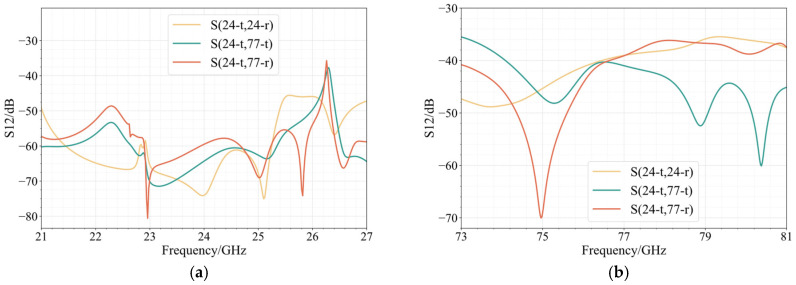
Simulation results of isolation between arrays: (**a**) 24 GHz band and (**b**) 77 GHz band.

**Figure 13 sensors-25-00481-f013:**
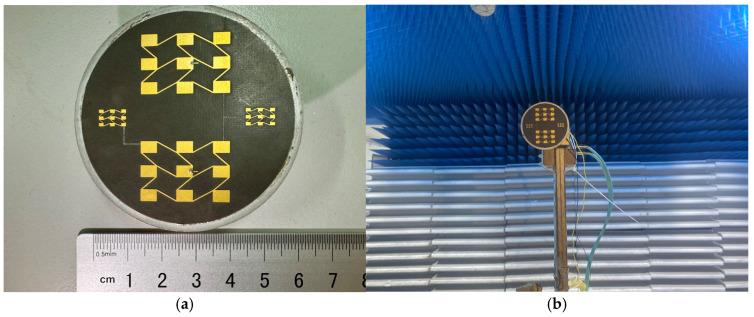
(**a**) Integrated array physical image. (**b**) Integrated array gain measurement environment.

**Figure 14 sensors-25-00481-f014:**
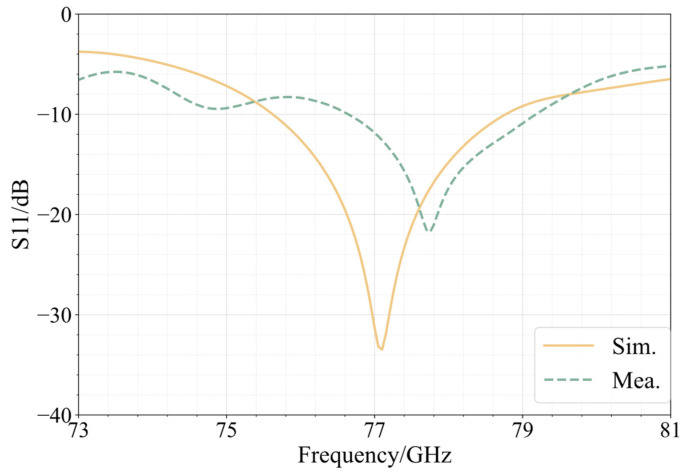
S11 simulation and measurement results of 77 GHz antenna in integrated array.

**Figure 15 sensors-25-00481-f015:**
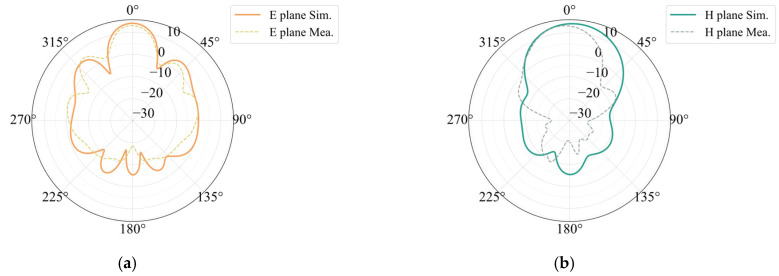
Gain simulation and measurement results of 77 GHz antenna in integrated array: (**a**) E-plane and (**b**) H-plane.

**Figure 16 sensors-25-00481-f016:**
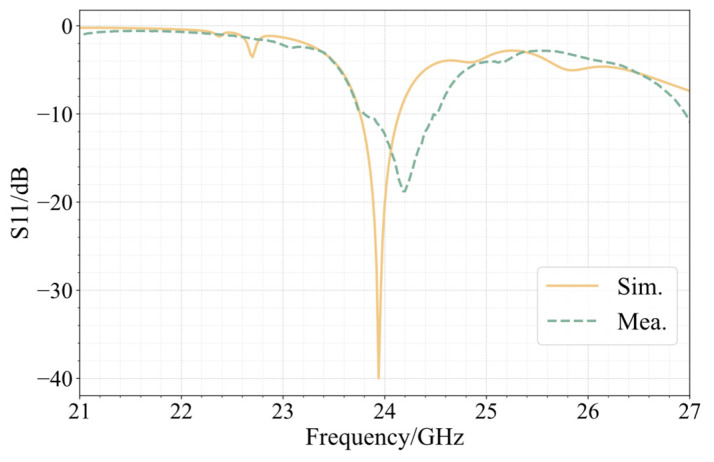
S11 simulation and measurement results of 24 GHz antenna in integrated array.

**Figure 17 sensors-25-00481-f017:**
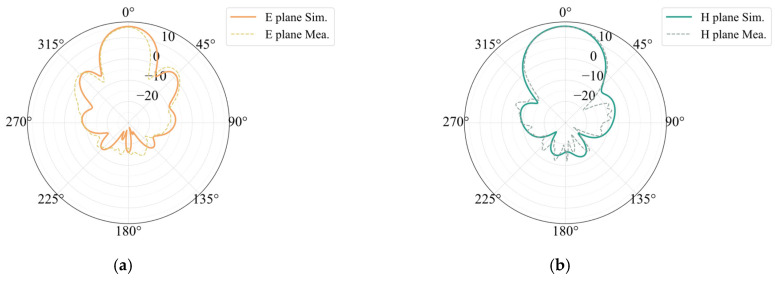
Gain simulation and measurement results of 24 GHz antenna in integrated array: (**a**) E-plane and (**b**) H-plane.

**Table 1 sensors-25-00481-t001:** Comparison between different antennas.

Antenna Type	Antenna Size (mm^2^)	Frequency (GHz)	Bandwidth (GHz)	Gain (dB)	Sidelobe Level (dB)	Cross-Polarization (dB)
Initial patch	1.12 × 1.14	77	4.04	7.43	-	−51.68 @ center feed −5.16 @ corner feed
Miniaturized patch	1.13 × 0.68	77	3.18	6.61	-	−52.16 dB @ center feed −20.03 @ corner feed
1 × 3 linear array	2.26 × 4.74	77	7.18	10.10	−14.29	−53.54
3 × 3 planar array	7.72 × 4.74	77	3.15	14.19	−9.14	−29.26
3 × 3 planar array	24.76 × 16.92	24	0.38	15.34	−12.85	−40.52
Circular array Ref [21]	23.78 × 23.78	7.95	6.3	8.4	-	−25
1 × 6 linear array Ref [22]	6 × 26	28.8	0.7	9.19	−2	-
1 × 2 linear array Ref [23]	9 × 10.3	9.42	1.43	9.42	-	-
1 × 4 linear array Ref [24]	11.3 × 22.5	30	1.25	12.63	−13.66	−44.79
1 × 4 linear array Ref [25]	31 × 10	28	1.29	13.5	−14.5	-

## Data Availability

All data are included within the manuscript.

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
