# Peer review of "Design of Microstrip Antenna Integrating 24 GHz and 77 GHz Compact High-Gain Arrays"

_sensors, 2025, doi:10.3390/s25020481_

Round 1
Reviewer 1 Report
Comments and Suggestions for Authors
The authors have presented a dual-band antenna array with high gain and compact size features. In fact, the contribution is significant. There are several comments as follows:
1. Introduction is too long, especially the first paragraphs.
2. The size miniaturization has no meaning. The authors just reduce the patch width, which has a trade-off with bandwidth. Antenna miniatuzation is a method to decrese the physical resonant length (l), not w.
3. In Fig. 2, adding the gain curve is better.
4. The gain enhancement of 3x3 (9 elements) is quite low compared to a single element (7.4 dBi). Why?
5. In Fig. 6, the grating lobe in E-plane is high. Whats the reason?
6. This paper lacks of key parameters studies, which is very important to fully understand the operating principle of the antenna. Pls add at least 3 key design parameters.
7. For integrated array, hows to choose the distance between the array elements?
Author Response
We have revised the manuscript based on the feedback we received. Attached please find the revised version, which we would like to submit for your kind consideration.

Reviewer 2 Report
Comments and Suggestions for Authors
The paper presents a study on the design of an integrated array antenna operating at 24 GHz and 77 GHz frequencies, addressing contemporary challenges of miniaturization and efficiency in radar systems. Although the study proposes innovative solutions, a critical analysis reveals both strengths and weaknesses in the methodology, results, and practical implications.
The strengths of this study lie in its innovative nature and promising performance. First, the antenna proposes a novel approach to width reduction, allowing a significant decrease in volume by 39.82%, a major asset for radar platforms where space is limited. The series corner feed simplifies the feed network, reducing energy losses while improving gain, thus outperforming more complex traditional designs. In terms of performance, the antenna displays high gains of 15.34 dBi at 24 GHz and 14.19 dBi at 77 GHz, with satisfactory levels of sidelobes and cross-polarization, meeting the requirements of modern radar systems. Finally, the versatility of the antenna is highlighted by its potential applications in various fields such as weather monitoring and automotive radars, demonstrating its suitability for diversified practical uses.
The weaknesses of this study mainly lie in some methodological limitations. First, a strong reliance on simulations, although supported by promising experimental results, may question the validity of the performances in real conditions. Further tests in operational environments would be necessary to confirm the reliability of the antenna. Furthermore, the reduction of the patch width, although effective in terms of miniaturization, could affect the antenna bandwidth, an aspect requiring further analysis. Finally, high-frequency miniaturization introduces technical challenges related to manufacturing precision, a point briefly touched upon in the report, but which would deserve a more detailed exploration to consider suitable solutions in an industrial setting.
In conclusion, the paper proposes a novel design of an integrated array antenna that meets the increasing needs for miniaturization and performance in radar systems. However, further studies are crucial to validate the experimental results and assess the practical implications of this design. Furthermore, special attention should be paid to the technical challenges related to manufacturing to ensure that this technology can be effectively implemented in real applications.
Author Response

(The authors gave the same response as above.)

Reviewer 3 Report
Comments and Suggestions for Authors
This paper proposes an integrated array antenna operating in the 24 GHz and 77 GHz frequency bands. The microstrip antenna array uses the width reduction approach and corner series feeding to reduce the volume of the array, eliminate the need for a complex feeding network, minimize feeding losses, and enhance the antenna's gain. Here are my comments:
1. The Introduction is a bit redundant and needs to be reorganized.
2. The innovation mentioned in the paper is to reduce the width of microstrip patches to reduce antenna size and achieve antenna miniaturization. However, in the millimeter wave frequency band, especially for the 77G frequency band, the antenna size is already very small, such as a width of w=0.68mm. Is it still necessary to design a miniaturized antenna?
3. The primary polarization component of the antenna's radiation pattern is mainly dependent on the electric field radiation from the narrow edge. But when the width W decreases, it directly reduces the effective length of radiation, resulting in a decrease in antenna gain, as shown in Figure 3, where the gain decreases from 7.43dBi to 6.61dBi. Therefore, reducing the patch width of microstrip antenna also has obvious disadvantages.
4. In many engineering applications, a microstrip antenna with a width smaller than its length is a commonly used design method and cannot be considered an obvious innovation.
5. Corner series feeding is an interesting feeding method. Please provide a more detailed analysis in the paper, such as the principle of improving antenna gain and its impact on the cross polarization.
6. Please compare the antenna size, bandwidth, gain, sidelobe, cross polarization, isolation, etc. of this paper with other antennas in some published paper to demonstrate the advantages of the proposed antenna.
7. Can the fabrication accuracy of the second decimal place in Figure 13 be guaranteed? For example, w=0.68mm, ly=1.79mm.
Author Response

(The authors gave the same response as above.)

Reviewer 4 Report
Comments and Suggestions for Authors
1,“Figures 3 and 4 present the simulation results of the antenna element” in page4 line 135, should be "Figures 2 and 3“
2、The method of angle feeding is not clearly described, some patches have three connection points, while others only have two. In addition, the patch has four corner points, and there seems to be no pattern in how to choose two or three of them for feeding or interconnecting with other patches
Author Response

(The authors gave the same response as above.)

Round 2
Reviewer 1 Report
Comments and Suggestions for Authors
The authors have well addressed all concerns.
Author Response
We appreciate your insightful and constructive feedback on this article.
Reviewer 3 Report
Comments and Suggestions for Authors
Please provide some explanation for Table 1 in the paper to demonstrate that the proposed antenna has performance advantages compared to the antennas in published references.
Author Response
In lines 285-305, the author describes and compares the antennas in Table 1, emphasizing the article's superior design.
Reviewer 4 Report
Comments and Suggestions for Authors
The microstrip antenna array proposed in this paper has the advantage of small size and has certain reference value。
Author Response
We appreciate your recognition.